# A survey of authors publishing in four megajournals

David J. Solomon

Michigan State University, E. Lansing, MI, USA

Corresponding author
David J. Solomon,
dsolomon@msu.edu

## ABSTRACT

**Aim**. To determine the characteristics of megajournal authors, the nature of the manuscripts they are submitting to these journals, factors influencing their decision to publish in a megajournal, sources of funding for article processing charges (APCs) or other fees and their likelihood of submitting to a megajournal in the future.
**Methods**. Web-based survey of 2,128 authors who recently published in BMJ Open, PeerJ, PLOS ONE or SAGE Open.
**Results**. The response rate ranged from 26% for BMJ Open to 47% for SAGE Open. The authors were international, largely academics who had recently published in both subscription and Open Access (OA) journals. Across journals about 25% of the articles were preliminary findings and just under half were resubmissions of manuscripts rejected by other journals. Editors from other BMJ journals and perhaps to a lesser extent SAGE and PLOS journals appear to be encouraging authors to submit manuscripts that were rejected by the editor's journals to a megajournal published by the same publisher. Quality of the journal and speed of the review process were important factors across all four journals. Impact factor was important for PLOS ONE authors but less so for BMJ Open authors, which also has an impact factor. The review criteria and the fact the journal was OA were other significant factors particularly important for PeerJ authors. The reputation of the publisher was an important factor for SAGE Open and BMJ Open. About half of PLOS ONE and around a third of BMJ Open and PeerJ authors used grant funding for publishing charges while only about 10% of SAGE Open used grant funding for publication charges. Around 60% of SAGE Open and 32% of PeerJ authors self-funded their publication fees however the fees are modest for these journals. The majority of authors from all 4 journals were pleased with their experience and indicated they were likely to submit to the same or similar journal in the future.
**Conclusions**. Megajournals are drawing an international group of authors who tend to be experienced academics. They are choosing to publish in megajournals for a variety of reasons but most seem to value the quality of the journal and the speed of the review/publication process. Having a broad scope was not a key factor for most authors though being OA was important for PeerJ and SAGE Open authors. Most authors appeared pleased with the experience and indicated they are likely to submit future manuscripts to the same or similar megajournal which seems to suggest these journals will continue to grow in popularity.

## INTRODUCTION

With the launch of PLOS ONE in 2007 and its phenomenal growth over the last 7 years, megajournals have become a major venue for disseminating scientific research. These journals are open access (OA), have broad scopes; a review process focusing exclusively on whether the methodology is sound and an accelerated publication process. There are now over 20 such journals including recently announced megajournals from the Royal Society, the American Association for the Advancement of Science and the American Educational Research Association. While the growth of PLOS ONE has moderated, the growth of megajournal publishing as a whole continues to be exponential and in total these journals are now publishing approximately 4,000 articles a month (*Binfield, 2013*) or an estimated 2.5% of the total scholarly articles published based on estimates from *Ware & Mabe (2012)*. Clearly many authors are finding this form of publication attractive.

Although author surveys have been conducted by megajournal publishers (*Patterson, 2011*; *PeerJ, 2013*), to my knowledge there has not been an independent survey of megajournal authors. As one might expect, publisher surveys have focused on the author experience with the submission, review and publication processes. While these issues are important, the factors that influence authors to choose to publish in a megajournal and the means by which they are able to fund the article processing charge (APC) or other fees are also important issues in understanding this new form of scientific publishing.

The Scientific, Technical and Medical (STM) publishing market alone is approaching 10 billion USD with about 73% of the 11,550 English language journals from non-profit publishers (*Ware & Mabe, 2012*). If authors find this new form of publishing attractive and begin publishing a significant portion of their manuscripts in megajournals, this could have a profound impact on scientific publishing for both societies and commercial publishers. The increasing number of major societies such as AAAS and AERA launching these journals highlights this potential shift in scholarly publishing.

The growth of the APC funded OA market itself is becoming disruptive to the publishing industry. The shift from funding publication via subscriptions to APCs used by the majority of megajournals shifts from readers or their proxy being the purchaser of publishing services to authors, their institutions and/or their funders being the purchaser. The factors that influence authors' choice of a journal and the sources of funding they have available for paying publishing charges becomes much more critical when publishing is funded through publishing fees. How authors feel about this issue has broad implications for maintaining a transparent, cost effective and competitive market for APC funded publication (*Björk & Solomon, 2014*). Understanding how authors are funding publishing fees and how these fees are impacting on their choice of where to publish their research is essential for understanding the economics of this funding model and the extent we are likely to see a transparent and reasonably priced scholarly publishing system if megajournals continue to gain an increasing share of the scholarly publishing market.

I conducted a web-based survey of corresponding authors who recently published articles in four megajournals. These included PLOS ONE, BMJ Open, SAGE Open and PeerJ. I chose PLOS ONE and BMJ Open because they were established biomedical

**Table 1  Journal description.**

| | Journal | | | |
| --- | --- | --- | --- | --- |
| | **BMJ Open** | **PeerJ** | **PLOS ONE** | **SAGE Open** |
| Discipline | Biomedicine | Biomedicine | Biomedicine | Social science |
| Start year | 2011 | 2013 | 2007 | 2011 |
| Article process charge | £1350 | $99–$299[a] | $1350 | $99 |
| 2012 impact factor | 1.58 | NA | 3.73 | NA |

**Notes.**

[a] One-time membership fee allowing a specified number of publications a year.

journals in operation long enough to receive an impact factor. SAGE Open is the only established megajournal in the social sciences and offered interesting cross-discipline comparison with the mostly biomedical journals that were launched after the success of PLOS ONE. PeerJ is a very innovative biomedical megajournal with a unique membership funding model and also offered an interesting comparison with the other three journals. Table 1 provides basic information on each journal. While there were about 20 megajournals in operation at the time this study was completed, I had limited resources and felt these four journals offered a representative picture of the authors publishing in this journal format.

The goals of the survey were describing the authors choosing to publish in these four megajournals; determining the factors influencing their decision to publish in the journal they chose and the means by which they funded the APC or in the case of PeerJ, the membership fee. I was also interested in whether their article was a resubmission after being rejected by another journal and whether these articles described preliminary research that and was the basis of what the authors planned to publish in more complete form in another venue. In other words were these journals containing largely preliminary work or manuscripts authors had difficulty publishing elsewhere. Authors were also asked the likelihood of submitting a manuscript to that journal or another similar megajournal. I would have like to ask a broader set of questions but felt it was important to keep the survey short in the interest of increasing response rates from the busy researchers publishing in these journals. As noted this new form of scientific publishing has grown quickly and new megajournals are rapidly being launched. Understanding the type of material that is being published, the motivations for authors choosing to publish in these journals and how they are funding author fees is essential for understanding the rapidly evolving scholarly publishing system. This information is useful for funding agencies, scholarly societies and libraries in adjusting to the rapidly evolving scholarly publishing ecosystem.

## METHODOLOGY

### Sample

I sampled authors who published in BMJ Open and SAGE Open during 2013. The journals had published around 800 and 300 articles respectively during 2013. Using software I developed, I "scraped" the corresponding author's name, email and the title of the article

off the publisher's web site. The software was run in mid-December 2013 and I attempted to scrape all the published articles starting in January 2013 up until the time the data were collected. PLOS ONE published over 30,000 articles during 2013, far more than would be reasonable to sample. I originally sampled about 1,000 authors whose articles were published during the first 2 weeks of December 2013 scraping the author contact information from the PLOS ONE web site using similar software to that used for BMC Open and SAGE Open. These were used to pilot the survey, software and email contact procedures. Due to changes in the survey based on the pilot results and an error in the software that failed to record one of the questions; these data were not used in the study. I sampled approximately 1,000 PLOS ONE authors whose articles were published in the last two weeks of January 2014. PeerJ provided me with equivalent author information for all the authors publishing in PeerJ from its launch in February 2013 through February 2014.

The email lists were stripped of duplicates and a few of the data records had names and other information garbled by inconsistencies in the character set during the scraping process. Where it was not possible to reconstruct the information, these authors were not used in the study. A total of 935, 728, 231 and 234 authors were surveyed from PLOS ONE, BMJ Open, SAGE Open and PeerJ respectively resulting in a grand total of 2,128 authors surveyed.

## Survey

I followed guidelines outlined by *Dillman (2000)* in an attempt to maximize survey response rates. Email requests to participate in the survey were addressed to authors by name with the title of their recently published article. A survey link was provided for completing the survey including a coded ID passed via the URL that identitied the author. While the survey data remained anonymous, authors completing the survey were flagged in the data table with their emails indicating they had completed the survey. Authors who did not complete the survey were sent a reminder email approximately one week after the original email. An example of the survey can be viewed at:

http://openaccesspublishing.org/survey/Example_Survey.html.

A slightly different version of the survey was used for the authors publishing in PeerJ to reflect the membership funding model as opposed to the APC funding model used by the other 3 journals. An example of the PeerJ survey can be viewed at:

http://openaccesspublishing.org/survey/Example_PJsurvey.html.

Where appropriate, chi-square tests were performed to assess the probability the differences in the percentages in the authors' responses among the journals were due to chance. Where the probability the differences were due to chance were less than $p < 0.01$ it is noted in the table legends.

The survey protocol was deemed exempt by the Michigan State University Biomedical and Health Institutional Review Board, IRB Number: x14-029e; i045267.

## RESULTS

Authors were given the opportunity to include written comments concerning their employment, choice of a journal and more general comments about their experience

publishing in the megajournal they chose. The comments in a PDF format and a copy of the data in Excel format are available at:

http://dx.doi.org/10.6084/m9.figshare.962954.

The software used to scrape the author names and email addresses and the email sent to the authors requesting their permission are also included at the URL above.

About 35 of the authors' emails were not valid. SAGE Open had the highest response rate at around 47%, followed by PeerJ at around 40% with substantially lower response rates for PLOS ONE and BMJ Open at 29% and 26% respectively. A total of 665 authors responded to the survey. These results are roughly consistent with a similar survey I conducted with a colleague that included OA journals charging APCs from a wide range of disciplines (*Solomon & Björk, 2012*).

Approximately 25% of the authors across all the journals and 40% of the authors of PeerJ and SAGE Open articles were from the USA. Researchers from United Kingdom and Australia published the highest percentage of articles in BMJ Open. Articles were most evenly distributed geographically in PLOS ONE but all four megajournals were very international with authors spanning the globe. A listing of the number of articles from each country for each of the four journals is contained in the supplemental materials.

Table 2 describes the positions held by the corresponding authors. A further description of the tenure status and rank of authors with academic positions is contained in the supplemental material. It should be noted that in many countries the concept of tenure and the way faulty ranking is done did not conform to the way it is categorized in the survey. The authors were allowed to mark multiple types of positions and so the percentages of authors marking each category add up to more than 100% and the counts add up to more than the total number of authors. Most authors, about 85% publishing in each journal, are academics. Over a fifth of the PLOS ONE authors worked in research laboratories as did 16% of the PeerJ authors. Much lower percentages of BMC Open and SAGE Open authors worked in laboratories. Roughly 10% of the authors worked for the government with the percentage being a little higher for BMJ Open. Only a very small percentage of the authors worked in private industry.

Authors were asked about their previous publishing history both in terms of subscription and open access journals. Tables 3A and 3B present these results. Most of the authors had at least some experience over the last few years publishing in both subscription and open access journals with PeerJ authors appearing to having slightly more experience publishing in OA journals as compared with the other authors though these differences did not reach statistical significance $p < 0.01$.

Authors were also asked whether their manuscript described preliminary findings from a study from which they planned to publish more complete data at a later date and whether the paper was a resubmission of a manuscript they had attempted to publish in another journal. Tables 4A and 4B present the results. About 10% of the manuscripts in PeerJ were described by their authors as preliminary findings while the percentage was more than double that in the other three journals. Two thirds of the articles in BMJ Open were resubmissions; slightly fewer than half the articles in PLOS ONE and SAGE

**Table 2 Types of positions held by corresponding authors.[a]**

| | | Journal | | | | |
|---|---|---|---|---|---|---|
| | | BMJ Open | PeerJ | PLOS ONE | SAGE Open | Total |
| Academic | Count | 156 | 81 | 233 | 92 | 562 |
| | Percent | 83.4% | 85.3% | 85.7% | 86.0% | |
| Research laboratory[*] | Count | 10 | 15 | 59 | 6 | 90 |
| | Percent | 5.3% | 15.8% | 21.7% | 5.6% | |
| Private industry | Count | 3 | 3 | 6 | 6 | 18 |
| | Percent | 1.6% | 3.2% | 2.2% | 5.6% | |
| Government | Count | 25 | 9 | 32 | 11 | 77 |
| | Percent | 13.4% | 9.5% | 11.8% | 10.3% | |
| Non-Profit | Count | 9 | 10 | 20 | 4 | 43 |
| | Percent | 4.8% | 10.5% | 7.4% | 3.7% | |
| Other | Count | 26 | 7 | 18 | 11 | 62 |
| | Percent | 13.9% | 7.4% | 6.6% | 10.3% | |
| Authors represented | Count | 187 | 95 | 272 | 107 | 661 |

**Notes.**
[a] Authors had the opportunity to mark more than one category so numbers total to more than the number of respondents. The percentages are for the numbers of authors from a specific journal marking that type of position.
[*] Differences among journals were statistically significant $p < 0.01$.

Open and about 37% in PeerJ were resubmissions. Comments from authors publishing in BMJ Open, PLOS ONE and SAGE Open indicated that in at least some of the cases, journal editors from the same publisher (BMJ, PLOS and SAGE) encouraged the authors to submit their manuscript to the publisher's megajournal after it was rejected in the original journal. These were most prevalent for BMJ where 11 of the authors indicated they were encouraged by the editor of another BMJ group journal to resubmit to BMJ Open. It should be noted that authors submitting to other BMJ journals can ask that their manuscripts be automatically considered by BMJ Open if not accepted by the other journal (*BMJ, 2014*). PLOS ONE's instructions for authors indicates the publisher will help in transferring manuscripts from one PLOS journal to another but encourage authors to carefully consider which PLOS journal would be most appropriate for their manuscript before submission.

Authors were asked to rate the importance of each of 10 factors in their decision to submit their paper to the megajournal in which they published. They also were asked to select the most important factor and were given the option of describing other factors in a textbox. As noted above, a slightly different survey was sent to PeerJ authors given the difference in funding model and that the journal has not yet been given an impact factor. Two of the 10 factors listed for this question differed. Instead of impact factor, authors were asked to rate the importance of the website, which is fairly innovative and instead of the amount of the APC they were asked to rate the importance of PeerJ's membership funding model. Importance was rated on a four point scale from not important to very important. Authors were also given the option of marking that the factor negatively influenced their

**Table 3  Number of peer reviewed publications last 3 years (subscription and open access).**

| | | Journal | | | | |
|---|---|---|---|---|---|---|
| | | BMJ | PeerJ | PLOS | Sage | Total |
| **A. Subscription** | | | | | | |
| None | Count | 15 | 8 | 30 | 22 | 75 |
| | % within Journal | 7.9% | 8.4% | 11.0% | 20.2% | 11.3% |
| 1–2 | Count | 28 | 15 | 42 | 20 | 105 |
| | % within Journal | 14.8% | 15.8% | 15.4% | 18.3% | 15.8% |
| 3–5 | Count | 46 | 15 | 79 | 40 | 180 |
| | % within Journal | 24.3% | 15.8% | 29.0% | 36.7% | 27.1% |
| 6–10 | Count | 34 | 20 | 47 | 14 | 115 |
| | % within Journal | 18.0% | 21.1% | 17.3% | 12.8% | 17.3% |
| Over 10 | Count | 66 | 37 | 74 | 13 | 190 |
| | % within Journal | 34.9% | 38.9% | 27.2% | 11.9% | 28.6% |
| | Count | 189 | 95 | 272 | 109 | 665 |
| | % within Journal | 100.0% | 100.0% | 100.0% | 100.0% | 100.0% |
| **B. Open access** | | | | | | |
| None | Count | 13 | 3 | 34 | 12 | 62 |
| | % within Journal | 6.9% | 3.2% | 12.5% | 11.0% | 9.3% |
| 1–2 | Count | 91 | 37 | 119 | 78 | 325 |
| | % within Journal | 48.1% | 38.9% | 43.8% | 71.6% | 48.9% |
| 3–5 | Count | 56 | 27 | 82 | 12 | 177 |
| | % within Journal | 29.6% | 28.4% | 30.1% | 11.0% | 26.6% |
| 6–10 | Count | 13 | 18 | 24 | 2 | 57 |
| | % within Journal | 6.9% | 18.9% | 8.8% | 1.8% | 8.6% |
| Over 10 | Count | 16 | 10 | 13 | 5 | 44 |
| | % within Journal | 8.5% | 10.5% | 4.8% | 4.6% | 6.6% |
| | Count | 189 | 95 | 272 | 109 | 665 |
| | % within Journal | 100.0% | 100.0% | 100.0% | 100.0% | 100.0% |

decision to submit. Table 5 presents the authors' selection of the most important factor, the average ratings on the four-point scale, and the number of authors indicating the factor negatively influenced their choice of the journal. Many authors failed to mark what they felt to be the most important factor but did rate the importance of the 10 factors.

The quality of the journals was a major consideration with over 20% of the authors that published in each journal rating it the most important consideration. The fact the journal was open access was the major consideration for PeerJ and SAGE Open authors but less of a consideration for PLOS ONE and BMJ Open authors. Just over 20% of the PLOS ONE authors rated the impact factor their most important consideration.

The authors were asked to list their sources of support for covering the APC or the membership fee in the case of PeerJ. They were given the option of listing multiple sources. The results are presented in Table 6. The individual percentages and counts add up to more than the totals reflecting that some authors listed multiple sources of funding. Two of the questions were altered for the PeerJ survey. "Institution funding based on an institutional

**Table 4  Does your paper describe preliminary findings or is it a resubmission?**

| | | Journal | | | | |
|---|---|---|---|---|---|---|
| | | BMJ Open | PeerJ | PLOS ONE | SAGE Open | Total |
| **A. Does your paper describe preliminary findings from a study from which you plan to publish more complete data at a later date?**[*] | | | | | | |
| No | Count | 146 | 85 | 191 | 81 | 503 |
| | % within Journal | 77.7% | 90.4% | 70.5% | 74.3% | 76.0% |
| Yes | Count | 42 | 9 | 80 | 28 | 159 |
| | % within Journal | 22.3% | 9.6% | 29.5% | 25.7% | 24.0% |
| | Count | 188 | 94 | 271 | 109 | 662 |
| | % within Journal | 100.0% | 100.0% | 100.0% | 100.0% | 100.0% |
| **B. Is this a resubmission of a manuscript you attempted to publish in another journal?**[*] | | | | | | |
| No | Count | 61 | 59 | 135 | 58 | 313 |
| | % within Journal | 32.4% | 62.8% | 50.2% | 53.2% | 47.4% |
| Yes | Count | 127 | 35 | 134 | 51 | 347 |
| | % within Journal | 67.6% | 37.2% | 49.8% | 46.8% | 52.6% |
| | Count | 188 | 94 | 269 | 109 | 660 |
| | % within Journal | 100.0% | 100.0% | 100.0% | 100.0% | 100.0% |

**Notes.**

[*] Differences among journals were statistically significant $p < 0.01$.

policy that funds open access publishing fees" was replaced with "Institution membership". "The fee was waived" was replaced with "promotional fee waiver".

Approximately half the authors publishing in PLOS ONE were able to use grant funding while roughly 35% and 30% of PeerJ and BMJ Open respectively used grant funding. Only 11% of SAGE Open authors used grant funding for the APC which is likely a reflection of the scope of the journal. The journals varied considerably in terms of the percentage of authors who self-funded the cost of publication. Almost 63% of SAGE Open authors used personal funds to pay the APC. It should be noted that SAGE Open reduced their APC to 99 USD in January 2013 (*SAGE, 2013*). Approximately one third of the PeerJ's authors indicated they used personal funds. Again, the membership fee for PeerJ is quite modest. Only about 11% and 8% of BMJ Open and PLOS ONE authors, respectively, used personal funds.

The authors were asked to rate on a five point scale how likely they would be to submit manuscripts to the same or similar megajournal in the future. Table 7 contains the breakdown by journal. The majority of authors across all journals indicated they would be likely to submit to the journal they published in or a similar megajournal in the future. Authors from PeerJ were most likely to resubmit with 72% indicating a 5 on the 5 point scale and another 20% a 4. SAGE Open authors were the least likely to submit again to the same or similar journal and yet almost 50% marked a 5 and another 27% a 4. It should be noted that the differences among the journals did not reach statistical significance $p < 0.01$. It appears that at least the authors who completed the survey were satisfied enough with the experience of publishing in a megajournal to seriously consider submitting to a megajournal in the future.

**Table 5** **The most important factors in choosing the journal and the importance of each factor in your decision to submit to this journal.**

**A. The most important factors in choosing the journal[*]**

| | | Journal | | | | |
| --- | --- | --- | --- | --- | --- | --- |
| | | BMJ Open | PeerJ | PLOS ONE | SAGE Open | Total |
| The quality of the journal | Percent | 27.8% | 20.6% | 27.7% | 20.6% | 25.7% |
| The impact factor of the journal | Percent | 13.5% | | 20.7% | 7.9% | 14.0% |
| Journal web site[a] | Percent | | 1.6% | | | |
| Reputation of the publisher | Percent | 18.3% | 3.2% | 2.7% | 17.5% | 9.4% |
| The Journal's audience | Percent | 11.1% | 4.8% | 10.3% | 4.8% | 8.9% |
| Having a broad scope | Percent | 0.8% | 0.0% | 4.3% | 6.3% | 3.0% |
| The speed of the review and publication process | Percent | 10.3% | 17.5% | 13.0% | 11.1% | 12.6% |
| The review criteria of the journal | Percent | 1.6% | 7.9% | 9.2% | 3.2% | 6.0% |
| Amount of the article processing charge | Percent | 1.6% | | .5% | 4.8% | 2.3% |
| Amount of the membership model[a] | | | 6.3% | | | |
| The fact the journal was Open Access | Percent | 8.7% | 28.6% | 10.3% | 22.2% | 14.2% |
| Recommendation of a colleague | Percent | 6.3% | 9.5% | 1.1% | 1.6% | 3.9% |
| | Count | 126 | 63 | 184 | 63 | 436 |

**B. The importance of each factor in your decision to submit to this journal[b]**

| | Journal | | | | | | | |
| --- | --- | --- | --- | --- | --- | --- | --- | --- |
| | BMJ Open | | PeerJ | | PLOS ONE | | SAGE Open | |
| | Mean | N | Mean | N | Mean | N | Mean | N |
| The quality of the journal[*] | 3.65 | 165 | 3.25 | 87 | 3.58 | (1) 238 | 3.36 | 98 |
| The impact factor of the journal[*] | 2.99 | (1) 176 | | | 3.32 | 247 | 2.74 | (4) 101 |
| Journal Web site[a] | | | 2.52 | 91 | | | | |
| Reputation of the publisher | 3.14 | 173 | 2.93 | 90 | 3.00 | (2) 253 | 3.31 | 98 |
| The Journal's audience[*] | 3.29 | 174 | 2.89 | 90 | 3.23 | (2) 249 | 3.11 | 100 |
| Having a broad scope | 2.67 | (1) 174 | 2.63 | 91 | 2.83 | 250 | 2.85 | (1) 100 |
| The speed of the review and publication process | 3.14 | 169 | 3.41 | 90 | 3.24 | (1) 244 | 3.23 | 95 |
| The review criteria of the journal | 2.91 | 175 | 3.27 | 92 | 3.09 | (1) 253 | 3.10 | (1) 101 |
| Amount of the article processing charge[*] | 2.23 | (14) 176 | | | 2.20 | (15) 255 | 2.71 | (3) 101 |
| Amount of the membership fee[a] | | | 2.72 | 88 | | | | |
| The fact the journal was Open Access[*] | 2.65 | (1) 173 | 3.30 | 89 | 2.78 | (1) 253 | 2.90 | (4) 99 |
| Recommendation of a colleague | 2.08 | (1) 170 | 2.13 | (1) 88 | 2.18 | (3) 252 | 1.86 | (3) 102 |

**Notes.**

[*] Differences among journals were statistically significant $p < 0.01$.

[a] Two questions differed in the survey taken by PeerJ authors.

[b] Ratings are on a 4 point scale from 1 = "Not Important" through 4 = "Very Important". Numbers in parentheses are the authors who marked the factor as negatively influencing their decision to publish in the journal.

The authors were given the option to write in general comments concerning their experience with the following prompt. "***Do you have any other thoughts about publishing your paper in [journal]?***" The comments, organized by journal, are available in the survey data archive at the URL above.
**Table 6  Sources of funding for publication fees.**

| | | Journal | | | | |
| --- | --- | --- | --- | --- | --- | --- |
| | | BMJ Open | PeerJ | PLOS ONE | SAGE Open | Total |
| Funds from a grant/contracting agency who funded the research/scholarship on which the article was based[*] | Count | 56 | 33 | 140 | 12 | 241 |
| | Percent | 30.3% | 35.5% | 52.0% | 11.2% | |
| Government funding based on a national policy that funds open access publishing fees[*] | Count | 12 | 0 | 41 | 1 | 54 |
| | Percent | 6.5% | 0.0% | 15.2% | .9% | |
| Institution funding based on an institutional policy that funds open access publishing fees | Count | 40 | | 48 | 11 | 99 |
| | Percent | 21.6% | | 17.8% | 10.3% | |
| Departmental or other institutional discretionary funds | Count | 48 | 13 | 53 | 15 | 129 |
| | Percent | 25.9% | 14.0% | 19.7% | 14.0% | |
| The fee was waived[*] | Count | 21 | | 12 | 1 | 34 |
| | Percent | 11.4% | | 4.5% | .9% | |
| Personal funds[*] | Count | 20 | 30 | 21 | 67 | 138 |
| | Percent | 10.8% | 32.3% | 7.8% | 62.6% | |
| Institution membership | Count | | 3 | | | 3 |
| | Percent | | 3.2% | | | |
| Promotional fee waiver | Count | | 25 | | | 25 |
| | Percent | | 26.9% | | | |
| Other | Count | 4 | 4 | 4 | 2 | 14 |
| | Percent | 2.2% | 4.3% | 1.5% | 1.9% | |
| | Count | 185 | 93 | 269 | 107 | 654 |

**Notes.**

[*] Differences among journals were statistically significant $p < 0.01$.

**Table 7  How likely would you be to submit future manuscripts to [journal name] or another similar megajournal in the future?**

| | | Journal | | | | Total |
| --- | --- | --- | --- | --- | --- | --- |
| | | BMJ Open | PeerJ | PLOS ONE | SAGE Open | |
| Very unlikely | Percent | 0.0% | 0.0% | .7% | 2.8% | .8% |
| | Percent | 4.3% | 2.2% | 3.0% | 2.8% | 3.2% |
| | Percent | 11.8% | 5.4% | 8.9% | 18.3% | 10.8% |
| | Percent | 26.7% | 20.4% | 26.4% | 26.6% | 25.7% |
| Very likely | Percent | 57.2% | 72.0% | 61.0% | 49.5% | 59.6% |
| | Count | 187 | 93 | 269 | 109 | 658 |

Comments that were clearly not informative, such as "thanks" or "no" were removed and a blank line was placed between each authors' comment. Otherwise the comments appear as received.

The comments across the four journals were in general quite positive reflecting the authors' indications they would likely publish in the journal they had or a similar mega-journal. The comments for BMJ Open and, in particular, PeerJ were extremely positive.

Eleven SAGE Open and 8 PLOS ONE authors complained about the length of the review process. In some cases based on the author's comments the journals had difficulty finding reviewers and/or academic editors with the appropriate expertise to review the submission. Several SAGE Open authors indicated they had review times of over a year. There were no such comments for BMJ Open and PeerJ. It should also be noted several authors were complementary of the SAGE Open and PLOS ONE review process. The APC was referenced in 18 of the comments from BMJ Open expressing a range of views.

## DISCUSSION

Both the number of megajournals and the articles they publish are growing rapidly. As this new format of scholarly publishing becomes established there is a need to gain a better understanding of the authors who are publishing in these journals, what is motivating them to choose this format and how are they covering the costs of publication. I believe this is the first systematic study exploring these issues.

This study has limitations that should be taken into account in interpreting the results. Chief among these is the potential of response bias. It is not clear how representative the respondents to the survey were of all authors publishing in these journals. Since the survey was designed to keep the responses anonymous, it is not possible to tell if the respondents differ in any systematic way from those who chose not to respond. In an attempt to increase the response rate, the survey was kept very short and hence the information that was captured is also fairly limited. In addition, scholarly articles, particularly in biomedicine tend to have several authors and it is not clear the views of the corresponding authors are representative of their coauthors. Only 4 journals were included in the study and may not represent megajournals in general. Also author opinions may evolve over time as this form of publishing become more established. Despite these and potentially other limitations the survey provides interesting information about authors' thoughts and experiences publishing in these four megajournals.

The journals appear to be attracting a very diverse group of researchers. Despite the fact BMJ Open and PLOS ONE have significant APCs, all four journals attract a very international set of authors. As one might expect the authors tend to be mainly academics. The authors as a whole appeared to be fairly experienced researchers who had a significant number of recent publications in both subscription and OA journals.

Authors were also asked if their study was a preliminary study and if it was a resubmission of an article originally sent to another journal. It would seem authors wishing to publish preliminary data from an ongoing study might find the megajournal format attractive given the review criteria are limited to the soundness of the methodology.

About a quarter of the authors in BMJ Open, PLOS ONE and SAGE Open indicated their manuscript did in fact describe preliminary results. Interestingly only about 10% of the authors publishing in PeerJ indicated this was the case. One might think PeerJ with its membership model and very rapid submission to first decision time would be attractive to authors publishing preliminary work but as yet, that does not seem to be a significant source of manuscripts for the journal.

These journals do seem to attract a significant number of resubmissions. As noted two thirds of the BMJ Open authors indicated their article was a resubmission of a manuscript rejected by another journal. There were spontaneous comments from 11 authors indicating they were encouraged to resubmit to BMJ Open after rejection by another BMJ journal. As noted above, BMJ does have a policy that upon request of the author, manuscripts not accepted by other BMJ journals will automatically be considered by BMJ Open. It appears this policy may account for many of the BMJ Open publications.

Just under half of the authors from PLOS ONE and SAGE Open indicated their publications were resubmissions. One author from each of these journals also spontaneously stated they were encouraged by the editor of another journal from the same publisher to submit to their megajournal after rejection. PeerJ had the lowest percentage of authors indicating their articles were resubmissions at 37%. While it may be an unrelated fact, PeerJ is not part of a larger publishing house. Since it is common for journals to have rejection rates well above 50%, it is not surprising many of the articles published in megajournals are resubmissions of manuscripts previously sent to another journal. It appears that to some extent publishers, most notably BMJ, are steering authors whose research may be sound but not of interest to their more selective journals to resubmit their manuscripts to their megajournal.

The quality of the journal and the speed of the review/publication process were the two factors rated highly across all four journals in the authors' decision to publish in the megajournal. The APC was a negative factor for a portion of the authors, particularly PLOS ONE and BMJ Open which have considerably higher fees. Being open access was one of the more important factors for PeerJ authors as well as SAGE Open authors but less so for the authors of the other two journals.

The impact factor was an important positive consideration for PLOS ONE and BMJ Open, the two journals that currently have an impact factor. *Davis (2014)* in a recent post on the Scholarly Kitchen blog considered the possibility that a decline in PLOS ONE submissions in February 2014 might be related to the decline in the journal's impact factor. A few PLOS ONE authors expressed concerns in the comments that PLOS ONE's impact factor appears to be declining. Since impact is such an important factor for a considerable number of authors publishing in PLOS ONE, a decline in impact could conceivably impact on submissions particularly if authors feel there is a likelihood of continued decline. It should be noted the vast majority of PLOS ONE authors indicated they were very likely to submit manuscripts to PLOS ONE or a similar journal in the future. Time will tell whether the February 2014 decline was an anomaly or will continue.

As one might expect, grants were the largest source of funding for APCs among the biomedical journals. It was far less common for SAGE Open authors where the vast majority of authors used personal funds. SAGE recently reduced the SAGE One APC to 99 USD, perhaps influenced by the fact their authors are largely self-funding publication and would naturally be very sensitive to the price of an APC. While considerably lower, around a third of the PeerJ authors paid their membership fee out of personal funds. Although a biomedical journal, PeerJ's membership fee is fairly low and may attract authors who are publishing unfunded research and might otherwise submit to a journal that does not require a publication fee. A large percentage of PLOS ONE and even more significantly BMJ Open authors used departmental/institutional discretionary funding or institutional funding based on a policy of funding APCs. About 25% of the SAGE Open authors also were able to fund publication with these sources. PeerJ has implemented institutional memberships with a number of universities and other organizations but very few of the authors who responded to the survey indicated they were able to take advantage of these agreements. PeerJ has had in place a fee waiver during a significant portion of the period data were collected. A little over a quarter of the authors were able to take advantage of publishing their manuscript without having to obtain a paid membership.

Authors are funding the cost of publishing in a variety of ways. As one might expect, covering a fee of around 1,350 USD or for some megajournals even higher is an issue for authors without grants or some form of institutional funding. Where it is possible to maintain APCs or membership fees in the range of 100 USD, it appears many authors are willing to pay the fees themselves. Given the relatively modest sample size I did not try to break down the data by country. While a fee of 100 USD might be affordable for researchers in the US or Western Europe, that may not be the case in other parts of the world.

Clearly the cost of publications fees is an issue for researchers in the social sciences and this pressure appears to have resulted in substantially lower publications fees for SAGE One. It seems that although a high APC was an issue for some of the respondents who published in PLOS ONE and BMJ Open these journals are attracting a large number of submissions and the pressure to keep publication fees low does not appear to be as great as in the social sciences one would assume because of the availability of funding either through granting agencies or the authors' institution.

The authors overwhelmingly indicated they would likely publish again in the same or another similar megajournal. Most of the comments were written were also positive. PeerJ authors seemed to be particularly satisfied with their publishing experience. As a new journal, the staff may be particularly sensitive to providing good service to the authors. As noted above, a few of the PLOS ONE and SAGE Open authors complained about exceptionally long review times. It appears that these journals occasionally experience difficulty locating appropriate review editors and/or reviewers. While this can be a challenge for any journal editor, having a very broad scope would likely increase the difficulty of finding qualified experts for peer review. Given the tremendous volume of PLOS ONE submissions it would not be surprising if they occasionally have difficulty finding appropriate review teams. Some authors were clearly frustrated with the length of

the review process. It seems most authors were reasonably satisfied given their willingness to submit to the journals or at least the journal format in the future.

What seems clear is the reputation of the journal and the quality of the service they provide are major factors in the choice of where these researchers published. For some megajournals and likely to a greater extent in some fields the impact factor as a proxy for quality is extremely important. Although journals have always competed for the best manuscripts, the APC funding model or a membership model such as used by PeerJ increases the importance of journals attracting submissions. Authors rather than readers or libraries as their proxy have become the customers with this business model. In this sense, one might expect an increasing focus by publishers on enhancing the services provided to authors such as fast efficient review and user friendly submission systems.

The impact on author fees appears to be more complex. In fields where funding for publishing is more readily available the pressure on publishers to lower APCs may be less or nearly nonexistent for journals with high impact factors and excellent reputations for quality. In fields such as social science where authors often self-fund publishing charges, price is an important factor in the decision where to publish and there appears to be real competitive pressure to keep author fees low. Grantors, universities and other agencies that provide funding for their researchers' publishing fees will need to find ways to balance subsidizing their researchers' publication fees while encouraging them to consider price as well as quality or impact in the choice of journals in which to publish. If not, there will likely be considerable inflation in publishing fees for the more prestigious journals in fields such as biomedicine where research funding is readily available.

Megajournals are becoming an established part of scientific publishing. The results of this survey seem to suggest that a broad range of authors are choosing to publish in this format and in general are satisfied with their experience.

### Funding
The author declares there was no funding for this work.

### Competing Interests
David Solomon is an Academic Editor for PeerJ.

### Author Contributions
- David J. Solomon conceived and designed the experiments, performed the experiments, analyzed the data, wrote the paper, prepared figures and/or tables, reviewed drafts of the paper.

### Human Ethics
The following information was supplied relating to ethical approvals (i.e., approving body and any reference numbers):

The survey protocol was deemed exempt by the Michigan State University Biomedical and Health Institutional Review Board, IRB Number: x14-029e; i045267.

### Data Deposition

The following information was supplied regarding the deposition of related data:

Solomon, David (2014): Data set for A Survey of Authors Publishing in 4 Mega-Journals.

Figshare http://dx.doi.org/10.6084/m9.figshare.962954.

### Supplemental Information

Supplemental information for this article can be found online at http://dx.doi.org/10.7717/peerj.365.

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
