# Peer review of "A survey of authors publishing in four megajournals"

_PeerJ, doi:10.7717/peerj.365_

## Round 0.1 · original submission · Minor Revisions

Thank you for your submission. As noted by one of the reviewers, I have exercised some judgement over whether it falls within the scope of PeerJ and I agree that it is of sufficient interest to our target audience.

The reviewers have provided a comprehensive critique of the work, which you should address point by point. I will not repeat them all here, but I would direct you to a few specific things:

(1) Ensure that you address the possible privacy concerns regarding the ability to identify participants from information in the supplementary material - especially the comments describing the authors' positions. If participants agreed to the use of all of their quotes, please say so, otherwise consider how you might anonymize them.

(2) It would aid readers to judge the validity of your sample collection strategy if you could make the web-scraping code available. Note also the comments about the "resampling" of PLOS ONE authors.

(3) The manuscript would benefit from a clearer statement of both its aims and conclusions.

(4) Please consider whether the data are amenable to a more comprehensive statistical treatment. This might support you in drawing some more specific conclusions (see (3)).

(5) One of the reviewers rightly pointed out the possible conflict of interest in PeerJ publishing an article, one of the conclusions of which is that PeerJ is perceived in a more positive light than other megajournals. It would be appropriate to include a statement to the effect that PeerJ did not solicit the article (I will raise this separately with the editorial office). In the interests of full disclosure, I serve as an academic editor for both PeerJ and PLOS ONE. This is entirely voluntary, and I do not derive any monetary benefit from the work, though I do have two PLOS ONE Editorial Board mugs.

(6) I noted the following typographic/grammatical errors:

L37 there -> their
L65 passed -> passing or that passed
L84 conducted _with_ a colleague
L92 in (first instance) -> is
L119 As noted above a slightly, different -> As noted above, a slightly different
L189 there -> their
L222 a third or one third
L223 membership _fee_
L223 _are_ publishing
L224 and APC -> an APC
Table 1 Awkward line break in Biomedicine under PLoS
Tables N.B. Correct form is now PLOS and should be PLOS ONE
Table 5a Leading zero missing
Table 5b marked the factor and negatively -> marked the factor as negatively

Note added by the Publisher: We can confirm that this article was not solicited by us. The emails (of published corresponding authors) which were supplied are publicly available via our site and so were simply supplied to prevent the need for scraping.

·

Basic reporting

There are potential privacy concerns: some of the comments listed in the supplementary material could lead to identification of individuals. There is no explicit mention that the study participants were asked to consent for their comments to be published. This is hopefully a minor oversight, but if participants didn't give this consent their free text comments should not be published.
In a few places the text isn't quite accurate or complete:
• In the last sentence of the Results it is stated that the comments from BMJ Open on the APC were "expressing a range of views". It would be more accurate to say that they mostly said that the APC was too high, as far as I can tell from the responses.
• It is stated that the SAGE Open APC was reduced to 99 USD - what was it before?
• It is stated that "BMJ does have a policy that upon request of the author, manuscripts not accepted by other BMJ journals will automatically be considered by BMJ Open." I am pretty sure this is also true for manuscripts submitted to other PLOS journals and PLOS ONE, and it may well be true for SAGE journals and SAGE Open too, so it is misleading to mention this just for BMJ journals.
The presentation of the alternative questions in surveys for PeerJ vs other journals is clear in Table 6 but less clear in Table 5. I suggest laying out Table 5 in the same way as Table 6, rather than lumping together impact factor and website in the currently confusing way.
The article would benefit from careful checking before publication as there are numerous typos and minor areas that are unclear. In particular, the following points seemed important:
• I think the word 'megajournal' should be written without a hyphen and as one word; the author could choose to have it hyphenated, but this should be consistent. Writing it as two words is definitely incorrect.
• The names of the four journals must be given in full throughout; BMJ Open should not be abbreviated to BMJ, as that is another journal, and PLOS One and SAGE Open should not be abbreviated to PLOS or SAGE, as that is the name of the publisher. These two abbreviations are used in the supplementary data files and in all the tables. BMJ Open is also called 'BMJ One' in one place.
• It should be stated somewhere in the Methods section how many authors completed the survey - I assume this is 665 as this is the total given for the tables.
• In Table 1 it should be stated what year the impact factors refer to (I assume 2012).
• 'Great Britain' should be 'UK' unless Northern Ireland was excluded. The data table has UK.
• "About 35 emails “bounced” due to bad addresses however I was unable to easily track the journal." - it is unclear what is meant in this sentence.
• "Comments that were clearly not informative, such as “thanks” or “no” were removed" but in fact a few such comments are still there.

Experimental design

Scope: this is not strictly speaking a biomedical article, but it probably fits within one of the 'General' subject areas covered by PeerJ.
The research question is not as well defined as I would like: it is 'To determine the characteristics of mega-journal authors', but it is not stated why it is useful to determine these characteristics. In the Introduction it is stated "the factors that influence authors to choose to publish in a mega journal and the means by which they are able to fund the article processing charge (APC) or other fees are also important issues in understanding this new form of scientific publishing" - again, it would be good to make it clear why exactly these are important to understand. The final concluding sentence of the article only mentions a broad range of authors and a generally good level of satisfaction, which doesn't make the main aim any clearer.
The author explains reasonably well why these four journals were chosen for the survey. However, it would be useful to mention others that could have been included, with reasons why they were not. Many are too new and still too small to have been considered, but Scientific Reports, SpringerPlus and Biology Direct come to mind as biomedical or general science megajournals that could have been included. The series that together form the equivalent of megajournals (BMC series, Hindawi journals, Frontiers journals) could also be mentioned and it could be explained why they weren't included.
Could any estimation be given of how random the sampling was? It is stated that there is response bias, but at the earlier stage in the experiment the sampling of PLOS One authors is not explained fully. How were the two sets of about 1000 authors from this journal selected? Exact numbers should be given in the Methods, rather than approximate ones. The word 'resample' may be being used wrongly in the second paragraph of the 'Sample' section - I would say the second set of 1000 PLOS ONE authors is a new sample, not a resampling.
Rather than asking authors how likely they would be to submit manuscripts to the same or similar mega journal in the future, it would have been nice to separate this out into their likelihood of submitting to that megajournal or (separately) another similar journal. This would make it clearer whether it was submission to a megajournal that was likely or submission to any particular journal. It might be worth discussing what difference this alternative question format might have made.
In the question on the types of position held by corresponding authors, it is difficult to interpret the results because the categories provided don’t seem to fit many of the authors' roles. Researchers in a clinical trial, or humanities researchers in a university, aren't covered by 'research laboratory' but are similar in many ways to those who work in a lab. This means that comparing medical and humanities journals with biomedical or general ones using this question is fairly meaningless. It is not clear whether 'academic' includes PhD students and postdocs or just those with a permanent position (or tenure as it is called in the US); also, in some countries in Europe researchers can be lecturers without tenure. It is probably useful to present this data but I don't find it very informative. If any cleaning up and reanalysis of this data can be done to clarify things that would be helpful.

Validity of the findings

Although all data is provided, the software developed by the author for scraping the journal websites to find author names is not provided. I suggest making this available either as a supplementary file or via a repository such as github. It is up to the journal whether this is essential for publication or not.
As mentioned in the previous section, the conclusions are not clearly stated. Some useful results have been obtained but some more specific conclusions from them would be nice.
No statistical analysis has been done on the data. I am not a statistician but I wonder whether statistical tests could be used to determine how likely it is that the differences between journals on particular criteria are to be the result of chance.

Additional comments

This is an interesting study on a relatively new kind of journal. It appears to be the first survey of megajournal authors that has been done except for those done by the journals themselves, and it is thus valuable. The results presented are supported by the data, which is all provided.
A few points could usefully be expanded or reworded in the Discussion:
• You mention "potentially other limitations". Could you say what other ones there might be? One mentioned above is that only four journals were studied, and only in one year; also that for one journal a sample of authors were used but for the others nearly all authors who published that year were surveyed (for good reasons).
• Regarding the PLOS ONE impact factor, you say "Time will tell whether the decline was an anomaly or will continue." Another possibility is that the first impact factor was the anomaly and that it will stabilize; this seems possible to me given the enthusiasm of authors and reviewers for new journals that can wane once they are established (this might also be one reason for the current enthusiasm for PeerJ).
• Calling 1350 USD an APC 'typical' for megajournals is not strictly accurate, as from figures I have collected from journal websites only three other megajournals have this APC (AIP Advances, Scientific Reports and Biology Open), some have lower and many have higher (BMC Series and Frontiers journals are all over 2000 USD). Perhaps saying that around 1500-2000 USD is typical would be better.
It would be useful to have some speculation in the Discussion about:
• The different response rates from the different journal authors
• what it might mean that PeerJ authors have more experience of publishing in OA journals than other journals' authors
• Why it might be that "It would seem authors wishing to publish preliminary data from an ongoing study might find the mega journal format attractive."
Finally, as one of the conclusions of this paper is that authors like PeerJ more than other megajournals, the journal itself might be seen as having a conflict of interest on this paper. I hope PeerJ will make efforts to ensure that this is mitigated, and perhaps a statement to that effect could be added.

·

Basic reporting

No comments

Experimental design

No comments

Validity of the findings

No comments

Additional comments

An interesting article shedding light on the authoral motivation for using mega-journals, establishing facts in a new area of investigation.

Minor comments/suggestions for corrections of typos (if I have understood the intention of the author correctly). I trust the author to amend the text as he sees necessary, this is merely suggestions.

Line 37 for [there article] substitute [their article]
Line 38 for [that and was the basis] substitute [that was the basis]
Line 81 [however I was unable to easily track the journal] I am unable to understand what the author means here?
Line 84 for [I conducted a colleague] substitute [I conducted with a colleague]
In the para at lines 77–85 the total number of responses and total response rate would have been a nice addition
Line 92 for [positions in contained] substitute [positions is contained]
Line 119 for [As noted above a slightly, different] substitute [As noted above, a slightly different]
Line 125 for [Table 6] substitute [Table 5/Tables 5a and 5b]
Line 135 for [7] substitude [6]
Line 143 for [authors using personal] substitute [authors used personal]
Line 171 for [what motivating] substitute [what motivates/what motivated or similar]
176 for [different] substitute [differ]

Table 5a, discussed in lines 117–127, has a markedly lower total of answers than what is found in the other tables. I find no information on why this is so. Could well deserve a comment.

---

## Round 0.2 · Minor Revisions

You have addressed the points raised in the original decision, however in doing so you seem to have introduced a few minor typographical errors. Please check the manuscript carefully and correct them.

Specifically, I noticed:

Page 1 Para 4 has a leading period (full stop)
Page 3 Para 8 "faulty ranking is none" should probably ready "faculty ranking is done"
Page 6 Para 3 "limted" -> "limited"
Page 7 Para 4 "results" -> "resulted"

This is not an exhaustive list and it might help to have a colleague to go over the final manuscript with a fresh eye.

---

## Round 0.3 · accepted · Accept

The reviewing PDF did not reflect the content of the Word document with tracked changes - I am basing my decision on the Word document and will flag the possible problem with the PDF with the editorial office.

Before publication, please check through one more time for typos, in particular:

-- Last paragraph on page 1, the sentence "Financial and other factors influencing authors’ choice of a journal in which to publish has..." should read "...have..."

-- Page 3 Check the wording of the sentence beginning "Where appropriate, chi-square tests were performed ..."

-- Also, please ensure that the abstract accurately reflects any changes you have made to the body of the manuscript (e.g. mega-journal -> megajournal, etc.).